# Cloud Update of Tiled Evidential Occupancy Grid Maps for the Multi-Vehicle Mapping

**DOI:** 10.3390/s18124119

**Published:** 2018-11-23

**Authors:** Kichun Jo, Sungjin Cho, Chansoo Kim, Paulo Resende, Benazouz Bradai, Fawzi Nashashibi, Myoungho Sunwoo

**Affiliations:** 1Department of Smart Vehicle Engineering, Konkuk university, Seoul 05029, Korea; kichun.jo@gmail.com; 2Department of Automotive Engineering, Hanyang university, Seoul 04763, Korea; sungjincho215@gmail.com (S.C.); chansoo7857@gmail.com (C.K.); 3Driving Assistance Research Center, Valeo, CEDEX 93012 Bobigny, France; paulo.resende@valeo.com (P.R.); benazouz.bradai@valeo.com (B.B.); 4Robotics and Intelligent Transportation Systems Team, INRIA Paris-Rocquencourt, 78153 Le Chesnay, France; fawzi.nashashibi@inria.fr

**Keywords:** Dempster-Shafer theory, occupancy grid map, cloud service, multi-vehicle mapping, LiDAR

## Abstract

Nowadays, many intelligent vehicles are equipped with various sensors to recognize their surrounding environment and to measure the motion or position of the vehicle. In addition, the number of intelligent vehicles equipped with a mobile Internet modem is increasing. Based on the sensors and Internet connection, the intelligent vehicles are able to share the sensor information with other vehicles via a cloud service. The sensor information sharing via the cloud service promises to improve the safe and efficient operation of the multiple intelligent vehicles. This paper presents a cloud update framework of occupancy grid maps for multiple intelligent vehicles in a large-scale environment. An evidential theory is applied to create the occupancy grid maps to address sensor disturbance such as measurement noise, occlusion and dynamic objects. Multiple vehicles equipped with LiDARs, motion sensors, and a low-cost GPS receiver create the evidential occupancy grid map (EOGM) for their passing trajectory based on GraphSLAM. A geodetic quad-tree tile system is applied to manage the EOGM, which provides a common tiling format to cover the large-scale environment. The created EOGM tiles are uploaded to EOGM cloud and merged with old EOGM tiles in the cloud using Dempster combination of evidential theory. Experiments were performed to evaluate the multiple EOGM mapping and the cloud update framework for large-scale road environment.

## 1. Introduction

Intelligent vehicles, such as self-driving cars and industrial mobile robots, require surrounding environment information to perform efficient and safe driving. A behavior decision system in the intelligent vehicle uses the environment information to determine the optimal behavior in consideration of passenger comfort and energy efficiency. The environment information is also used to find the safest vehicle motion that avoids collisions with obstacles [1,2,3,4]. An occupancy grid map is one way to model the driving environment by dividing space into grid (or voxel) cells and informing the occupancy state of each cell. The occupied states indicate whether the physical space of the cell is occupied or free of static objects.

Probabilistic approaches have been applied to infer the occupancy state of grid cells with consideration of sensor noise characteristics [5,6,7].These papers provide the probabilistic grid mapping using a Bayesian approach using a LiDAR beam model. Each cell state of the probabilistic occupancy grid map (POGM) models the posterior probability of occupancy. Although the probabilistic approaches of POGM are able to manage the sensor noise well, there are limits to conditions of sensor blind spots and dynamic objects. For the sensor blind spots, the occupancy state is *unknown* because of no sensor measurement for the detection area. For the dynamic objects, the occupancy state is *conflict* because the sensor measurement changes over time. In the probabilistic approach, the *unknown* and *conflict* state are only represented by a probability of 0.5, which is not clear to represent the state of the cell.

To overcome the limits of the POGM, evidential (Dempster-Shafer) theory has been applied to model the occupancy grid map [8,9,10]. The evidential framework can explicitly represent each cell as the *occupied*, *free*, *unknown* and *conflict* state. There are many previous studies to introduce the Dempster-Shafer approach for modeling of evidential occupancy grid map (EOGM) [11,12]. Karaman, et al. compared the performance of various types of approaches(including Dempster-Shafer approach) for environment modeling, and Cao, et al. described the 2D grid map building process using Demspter-Shafer evidence theory. In this literature, simple laser beam model and Demspter-Shafer approach based information fusion for map building were provided. To create the EOGMs for the surrounding environment of the vehicle, SLAM (Simultaneous Localization And Mapping) is applied basically. Celmens, et al. applied the evidential theory to SLAM application. This paper evaluated the evidential theory based SLAM with collision-free path planning. Evidential FastSLAM based on Rao-Brackwellized particle filter was used to build the EOGM and evaluated by comparing with POGM in [13]. Credibilist SLAM simultaneously estimates the vehicle pose and builds the EOGM using LiDAR [14,15]. This paper applies GraphSLAM which is a state-of-the-art method to obtain the pose of a vehicle based on the graph optimization technique [16].

The intelligent vehicles equipped with perception sensors (such as LiDAR and Stereo camera), GPS receiver, and motion sensors are able to create the EOGM for their passing trajectory. If these vehicles have a connection to the Internet, the created EOGMs can be shared with other vehicles through a cloud map service. The cloud sharing of the multi-vehicle EOGMs provides many benefits to operate intelligent vehicles [2,17,18,19]. The intelligent vehicle can plan the global behaviors and motions by reflecting the distant area’s EOGM downloaded from the cloud. The accuracy and reliability of EOGM in the cloud will be improved by merging the multiple EOGMs from several intelligent vehicles. Also, vehicles without perception sensors can use the EOGM generated by other vehicles if they only have an Internet connection.

For the cloud sharing of multi-vehicle EOGMs, there are two technical issues to be addressed. First, the EOGM must have a standard data format to be shared with other vehicles and to deal with the large-scale area. In the small area such as an indoor space, the EOGM is able to be implemented with single rectangular matrix data structure. However, the single rectangular matrix is not efficient format to cover the large-scale area for multiple vehicles. Second, evidential update method of EOGM is required to merge the new EOGMs updated from multiple on-road vehicles and old EOGM in the cloud. The evidential update also must take into account the aging factor of the old EOGM.

The main contribution of this paper is to propose a cloud update framework of evidential occupancy grid maps (EOGMs) for multiple intelligent vehicles in the large-scale real road environment. Each intelligent vehicle creates the EOGM for their moving trajectory based on GraphSLAM using LiDARs, motion sensors, and low-cost GPS receiver. The created EOGMs are uploaded and merged to EOGM in the map cloud. A standard tiling method of geodetic quad-tree tile system is applied to manage the EOGMs. The tiling method provides a common interface and data format to manage the large-scale EOGMs. Dempster combination rule of the evidential theory provides a theoretical basis to merge the new EOGM tiles to old EOGM tiles with consideration of the aging factor of old tiles. Experiments were performed to evaluate the cloud update framework of multi-vehicle EOGMs in the large-scale area.

This paper is organized as follows. Section 2 presents overall cloud update framework of multiple vehicle EOGMs. Section 3 describes the EOGM mapping based on the GraphSLAM. Section 4 describes the cloud update of EOGM based on a geodetic quad-tree tile system. Section 5 provides experimental results of the proposed cloud update framework. The final section provides conclusion and future works.

## 2. Cloud Update Framework for Multi-Vehicle

Figure 1 shows the overall framework for the EOGM cloud update of multi-vehicle mapping. The static objects for the driving environment are mapped into EOGM. Since the EOGM is based on the evidential theory, it is robust to the disturbance of dynamic objects such as moving vehicles, pedestrians, and motorcycles. The large-scale EOGM is split into local tiles based on a geodetic quad-tree tile system. The geodetic quad-tree tile system provides a standard tiling method for the large-scale environment. The cloud update framework consists of two steps. The first step is multi-vehicle mapping. The multiple vehicles create EOGM tiles for their driving trajectory. Each vehicle must be equipped with perception sensor (such as LiDAR), motion sensor, and low-cost GPS receiver for the EOGM tiles generation. GraphSLAM is used to create the optimal EOGM tile of the passing trajectory when the vehicle is out of the tile boundary. For the initialization of GraphSLAM, the position from low-cost GPS receiver and EOGM tiles downloaded from the cloud is used. Section 3 describes the details of the GraphSLAM-based EOGM mapping. The second step is cloud update of the EOGM tiles from multi-vehicle. The multiple vehicles upload their created EOGM tiles when the generation is complete and the network to cloud is available. The uploaded EOGM tiles are merged into the old EOGM tiles in the cloud. Since the EOGM tiles follow the standard tiling method of the geodetic quad-tree tile system, spatial matching between the uploaded EOGM tiles and the old EOGM tiles in the cloud is straightforward. The evidential framework of EOGM provides a theoretical basis to merge the new EOGM tiles to old EOGM tiles with taking into account the aging factor of the old tiles. Section 4 describes the details of the cloud update of EOGM tiles. Based on the iteration of the two-step framework for the EOGM cloud update, the EOGMs in the cloud keeps up-to-date with the change of the static environment.

## 3. Mapping of Evidential Occupancy Grid Map

### 3.1. Evidential Occupancy Grid Map (EOGM)

An occupancy grid map (OGM) models the driving space using discrete grids or voxels. This paper uses two-dimensional (2D) grid map to model the driving space due to the constraint of network bandwidth and storage space, but the same process of 2D mapping can be extended to three-dimensional (3D) voxel map. The unit of the discrete space is a cell. The cell contains information about whether the corresponding space is occupied by static obstacles. Probabilistic approaches can be applied to infer the cell occupancy. The probability of a cell is *zero* when the cell is a *free* state, whereas the probability is *one* when the cell is an *occupied* state. However, the probability cannot deal with the situation of *unknown* (no sensor information for the cell due to blind spot) and *conflict* (different sensor data for the same cell due to moving objects). In the probabilistic approach, the cell probability of both situations is set to 0.5, which is ambiguous probability to represent the *unknown* and *conflict*.

To handle the unknown and conflict state, an evidential approach based on Dempster-Shafer (DS) is applied to build an evidential occupancy grid map (EOGM). The cell of OGM has to consider the two states of *free* (*F*) and *occupied* (*O*) as a frame of discernment Ω={F,O}. In the EOGM based on the DS theory, the states of the cell can be extended to the power set $2^{\Omega }=\left\{F,O,\Omega ,\emptyset \right\}$, which is the set of all subsets of the Ω={F,O}. This means that the EOGM can consider two more cell states of Ω and ∅ than the probabilistic occupancy grid map (POGM). For elements of the power set, a mass function *m* can be used to quantify the evidence of the hypothesis. The mass of m(F) and m(O) represent the evidence of the cell is free and occupied, respectively. The mass of m(Ω) that considers the union of *F* and *O* represents the cell is *unknown*, and m(∅) represents the cell is *conflicted* by a different source of information. Based on the definition of the mass function in the evidential framework, the sum of mass functions for the power set must be *one*.

### 3.2. EOGM Mapping

The four masses of each cell in the EOGM can be determined based on the sensing information of time-of-flight (TOF) range sensors, such as LiDAR, radar, and ultrasonic sensors. Before the mapping process, the sensing information reflected by ground is eliminated by pre-processing. This paper uses the LiDAR as a representative of the TOF sensor. The EOGM mapping process consists of two steps: local EOGM mapping using LiDAR sensor model and global EOGM mapping based on the accumulation of the local EOGMs.

#### 3.2.1. Generation of Local EOGM Using LiDAR

The local EOGM represents an instant evidential grid map generated by a single scan of the LiDAR. Due to the scanning mechanism of the LiDAR, the local EOGM can be modeled as the polar grid map having the origin at the center of the LiDAR, as shown in Figure 2 [14]. In the angular sector (a) of Figure 2, the red cells on the polar grid map indicate the occupied states, which can be inferred by the echo of LiDAR beam against potential obstacles. The array of green cells before the occupied cell in distance axis indicates the state of free, and the array of blue cells after the occupied cell represents the unknown state. For the multiple echoes in one angular sector (Figure 2b), all the echoes of LiDAR set the cell as occupied, but the only array of cells located before the occupied cell that has minimum distance is set by free, and the remaining cells are set to the unknown state. The state of cells for the sensor model can be used to determine the initial value of mass functions for the certain cell {r,θ}, denoted by $\left[m_{L\{r,\theta \}}\left(F\right),m_{L\{r,\theta \}}\left(O\right),m_{L\{r,\theta \}}\left(\Omega \right),m_{L\{r,\theta \}}\left(\emptyset \right)\right]$, using
(1)$$\begin{array}{ll} m_{L\{r,\theta \}}\left(O\right) & =\left\{\begin{array}{cc} \lambda , & \mathrm{if}\;occluded\\ 0, & \mathrm{otherwise}\; \end{array}\right.\\ m_{L\{r,\theta \}}\left(F\right) & =\left\{\begin{array}{cc} \lambda , & \mathrm{if}\;free\\ 0, & \mathrm{otherwise}\; \end{array}\right.\\ m_{L\{r,\theta \}}\left(\Omega \right) & =\left\{\begin{array}{cc} 1, & \mathrm{if}\;unknown\\ 1-\lambda , & \mathrm{otherwise}\; \end{array}\right.\\ m_{L\{r,\theta \}}\left(\emptyset \right) & =0 \end{array}$$
where λ represents the confidence coefficient of the LiDAR [0 1].

#### 3.2.2. Global EOGM Generation

Since the local EOGM is temporal information about one scan of LiDAR and contains noise from moving objects, the local EOGM cannot be used as an environment model to represent a drivable space and static obstacle. The local EOGM should be incrementally integrated to the global EOGM to stably provide the drivable space and static obstacle with removing the moving object disturbance, as shown in Figure 3. The global EOGM is represented by the grid plane of Cartesian coordinates.

In the beginning, all cells in the global EOGM are initialized by the vacuous mass function
(2)$$\begin{array}{c} m_{G\{i,j\}}\left(F\right)=0,m_{G\{i,j\}}\left(O\right)=0\\ m_{G\{i,j\}}\left(\Omega \right)=1,m_{G\{i,j\}}\left(\emptyset \right)=0 \end{array}$$
that represents the no prior information. After the initialization, the local EOGMs of successive LiDAR scans are incrementally merged into the global EOGM. For the merging of a local EOGM to global EOGM, the local polar EOGM is converted to the local Cartesian EOGM which has the same grid configuration with global EOGM. The converted local EOGM in the Cartesian coordinates is then translated and rotated to align to the corresponding global EOGM by using the global pose (position and heading) from the GraphSLAM with the LiDAR extrinsic calibration parameters. The LiDAR extrinsic parameters are calibrated before the mapping process.

The mass function of each cell in the converted local EOGM can be represented by $m_{L\{i,j\}}$ that has the same index (i,j) with the global EOGM. Dempster combination rule of evidence theory is used to merge the local EOGM $m_{L\{i,j\},t}$ at time *t* with the global EOGM $m_{G\{i,j\},t-1}$ at time t−1:
(3)$$m_{G\{i,j\},t}=m_{G\{i,j\},t-1}⊕m_{L\{i,j\},t}.$$

The Dempster combination rule consists of the conjunctive combination rule (Equation 4) and the Dempster normalization (Equation 5) [20].
(4)$$\forall A\subseteq \Omega ,m_{1\cap 2}\left(A\right)=\sum_{B\cap C=A|B,C\subseteq \Omega }m_{1}\left(B\right)\cdot m_{2}\left(C\right)$$
(5)$$\begin{array}{cc} m_{1⊕2}\left(A\right) & =\frac{m_{1\cap 2}\left(A\right)}{1-m_{1\cap 2}\left(\emptyset \right)},\forall A\subseteq \Omega ,A\neq \emptyset \\ m_{1⊕2}\left(\emptyset \right) & =0 \end{array}$$

The $m_{1\cap 2}\left(A\right)$ means the result of conjunctive combination of $m_{1}$ and $m_{2}$ for state *A*, and $m_{1⊕2}\left(A\right)$ is the result from Demspter combination.

### 3.3. GraphSLAM

To generate the global EOGM by incrementally integrating the local EOGMs, the pose (position and heading) of a vehicle must be known. The state-of-the-art method to obtain the pose of a vehicle is GraphSLAM based on the graph optimization technique [16]. The SLAM problem can be represented using a graph (Figure 4) consisted of nodes that represent random variables of stochastic process and edges which represent constraints between the nodes.

The nodes of $x_{1:t}=\left[x_{1},\cdots ,x_{t}\right]$ represents a sequence of the poses for the discrete time steps 1 to *t*. The state of pose is composed of latitude, longitude, and heading. Low-cost GPS data is used to initialize and bound the state of poses. The map node *m* represents the static environment that is available to be recognized by perception sensors. The global EOGM is applied as the map node. The map node is initialized by downloaded EOGM from the cloud. The nodes of $u_{1:t}=\left[u_{1},\cdots ,u_{t}\right]$ represent a sequence of the vehicle motion (such as the yaw rate and speed) for the corresponding times of the states. The nodes of $z_{1:t}=\left[z_{1},\cdots ,z_{t}\right]$ represent the local EOGM measurement generated by the local EOGM mapping of LiDAR point cloud in the Section 3.2.1.

The edges are composed of a state transition model (blue arc) and a measurement model (green arc), as shown in the Figure 4. The state transition model $g(u_{t},x_{t-1})$ predicts the present pose $x_{t}$ from the previous pose $x_{t-1}$ with the input $u_{t}$. A constant velocity model is used for the state transition model with the motion inputs of the vehicle speed and yaw rate. We assumes the error $\epsilon_{u,t}$ between the pose $x_{t}$ and the predicted pose $g(u_{t},x_{t-1})$ follows the Gaussian distribution with covariance $P_{u}$. Edge constraint of the state transition model is represented by a quadratic form of the transition error $\epsilon_{u,t}$ and covariance $P_{u}$, as described in (7).
(6)$$\begin{array}{c} x_{t}=g\left(u_{t},x_{t-1}\right)+\epsilon_{u,t} \end{array}$$
(7)$$\begin{array}{c} {\left(x_{t}-g\left(u_{t},x_{t-1}\right)\right)}^{T}P_{u}^{-1}\left(x_{t}-g\left(u_{t},x_{t-1}\right)\right) \end{array}$$

The measurement model $h(x_{t},m)$ estimates the local EOGM measurement $z_{t}$ based on the pose $x_{t}$ with the global EOGM *m*, as described in (Equation 8). To construct the edge constraint of the measurement model, the error $\epsilon_{z,t}$ between the measurement $z_{t}$ and measurement model $h(x_{t},m)$ must be evaluated. However, since the $z_{t}$ and $h(x_{t},m)$ are EOGM data type, the error $\epsilon_{z,t}$ cannot be directly calculated with an analytical function. Instead of evaluating the error $\epsilon_{z,t}$, the similarity evaluation between two EOGMs based on the evidential scoring function is applied to construct the measurement model constraint. The evidential scoring function estimates the degree of matching of two EOGMs based on the sum of the *Occupied* masses in all cells of merged EOGM, as described in (9). The merged EOGM is obtained by applying the conjunctive combination rule (4) for the two EOGMs, $m_{1}=z_{t}$ and $m_{2}=h\left(x_{t},m\right)$. Based on the scoring function, the best matching pose $x_{t}^{\prime }$ which has the maximum score is evaluated, as described in (10). The best matching pose $x_{t}^{\prime }$ is used to construct the edge constraint with the pose $x_{t}$, as described in (11). The error between the pose $x_{t}$ and the best matching pose $x_{t}^{\prime }$ is assumed to follows the Gaussian distribution with covariance $P_{z}$.
(8)$$\begin{array}{c} z_{t}=h\left(x_{t},m\right)+\epsilon_{z,t} \end{array}$$
(9)$$\begin{array}{c} scoring\left(m_{1}=z_{t},m_{2}=h\left(x_{t},m\right)\right)=\sum_{\forall cells}m_{1\cap 2}\left(O\right) \end{array}$$
(10)$$\begin{array}{c} x_{t}^{\prime }=\underset{x}{\mathrm{argmax}}\;scoring\left(m_{1},m_{2}\right) \end{array}$$
(11)$$\begin{array}{c} {\left(x_{t}-x_{t}^{\prime }\right)}^{T}P_{z}^{-1}\left(x_{t}-x_{t}^{\prime }\right) \end{array}$$

A cost equation (12) is derived by sum of the log-likelihood constraints (7) and (11). When the cost function is minimized, the entire poses $x_{1:t}=\left[x_{1},\cdots ,x_{t}\right]$ are optimized. The cost optimization problem is solved by g2o library, which is open source c++ framework to optimize the nonlinear least squares problems [21].
(12)$$\begin{array}{cc} J & =\sum_{t}{\left(x_{t}-g\left(u_{t},x_{t-1}\right)\right)}^{T}P_{u}^{-1}\left(x_{t}-g\left(u_{t},x_{t-1}\right)\right)\\ & +\sum_{t}{\left(x_{t}-x_{t}^{\prime }\right)}^{T}P_{z}^{-1}\left(x_{t}-x_{t}^{\prime }\right) \end{array}$$

## 4. Cloud Update of EOGMs

We can create the global EOGM for small areas (less than one kilometer) using single rectangular matrix. However, the single rectangular matrix has limits to represent the global EOGM for the large-scale environment. First, the rectangular matrix of global EOGM causes wasting of memory space, as shown in Figure 3. Second, coordinate conversion between the geodetic coordinate and EOGM Cartesian coordinate for large-scale area causes a conversion approximation error. The global EOGM is based on a 2D Cartesian coordinate (east and north axis with meter unit), whereas the position for large areas on the Earth is normally represented by the geodetic coordinates (latitude and longitude). For the mapping of global EOGM in large areas, the mapping position in the geodetic coordinate such as WGS84 (World Geodetic System 1984) must transform into a Cartesian coordinate of global EOGM in order to be integrating the local EOGM. The coordinate conversion from geodetic coordinate to Cartesian coordinate is performed based on the tangential plane approximation for one reference point, as shown in Figure 5a. However, when the distance between the reference point and vehicle position is larger than 10 km, the approximation error of the coordinate conversion causes a distortion of global EOGM mapping [22].

To overcome the problems, a previous study presented a method to manage the grid map by tiles that split the space into many small areas [23]. However, since this tiling method can have a different configuration depending on the implementation of grid map, it is difficult to share or reuse the grid map by other vehicles. For the standardization of data structure of the global EOGM, this paper applies a geodetic quad-tree tile system to the global EOGM management. The geodetic quad-tree tile system is widely used for quick access and management of map in the digital map industry, such as Bing, HERE and Google map.

### 4.1. Geodetic Quad-Tree Tile System

The quad-tree tile system provides a common tile structure that can be used all around the world. The world can be split into hierarchical levels of tiles, which an upper-level tile is divided into four lower level tiles, as shown in Figure 6. The entire world is level 0, and the tiles of level 1 are divided by four regions of the level 0 tile, as shown in Figure 6a. The top two tiles of the level 1 are empty to keep the angular range of geodetic coordinate system (latitude: $\pm 90^{\circ }$, longitude: $\pm 180^{\circ }$) and to be the same angular size for each tile as (Equation 13).
(13)$$Degree\;of\;width\;and\;height\;per\;tile=360^{\circ }/2^{tile\;level}$$

The quad-key is a unique identification number for each tile. This number is assigned sequentially from the bottom left tile as shown in Figure 6a. The tiles of level two are also split into four tiles from one tile of level one tile. Figure 6b shows the level two tiles divided from the level one tile that contains a star mark (position of Eiffel tower). The quad-key of the level two tiles starts with the quad-key of the parent tile, and then the identification numbers are added, as shown in Figure 6b. By repeating this process, we can reach the level of tiles of the reasonable size for the global EOGM. Figure 6d shows tiles of level 16, which has latitude and longitude sizes of about $0.0055^{\circ }$ (width: 403.1 m, height: 610.88 m) with a quad-key 1220002130322231.

### 4.2. Geodetic Quad-Tree Tiling of Global EOGM

By assigning global EOGMs to the tiles of a certain level, we can manage the group of global EOGMs that can cover all around the world. To assign a global EOGM to a tile, the geodetic coordinate of the tile should be converted into Cartesian coordinate. For the coordinate conversion, the lower left corner point of the tile is set to be an origin of Cartesian coordinate that has east and north axis. Using the coordinate conversion based on the origin point, the tile size and certain position in the geodetic coordinate can be converted into the size and position (east and north) on the Cartesian coordinate. Figure 6e shows an example of coordinate conversion for a level 16 tile contained Eiffel tower. The lower left corner of the tile is set to origin of the Cartesian coordinate. The size ($360^{\circ }/2^{16}$) of the tile in geodetic coordinate is converted into size (403.1 m, 610.88 m) of Cartesian coordinate. The position of Eiffel tower ($48.8582^{\circ }$, $2.2947^{\circ }$) in the geodetic coordinate is converted to the position (297.24 m, 222.28 m) in the Cartesian coordinate of level 16 tile. In the same way, we can create and access the global EOGMs for the specific level tiles in the worlds.

There are many advantages when we use the quad-tree tile as the management system of the global EOGMs. First, the quad-tree tile representation can reduce the plane approximation error of coordinate conversion because the conversion is performed based on each origin point of the tiles, as shown in Figure 5b. Second, it is very effective in terms of memory utilization because only the global EOGM in the area of interest with the certain tile size need to be created and loaded into the memory. Figure 7 shows an example of the geodetic quad-tree tile-based global EOGM, which has same local mapping configuration with the Figure 3. Compared to Figure 3 with one large rectangular global EOGM, the quad-tree tile based management is more efficient for the memory utilization because the global EOGM for areas of no interest do not need to be updated. Third, since all quad-tree tiles have a unique quad-key, the searching and data management of the global EOGM for cloud update is straightforward. Finally, the most important thing is the quad-tree tile system provides the common interface and data format for the global EOGM, which is compatible for all around the world. Based on the common interface and format, multiple vehicles around the world can share the global EOGM for the environment perception.

### 4.3. Cloud Update of EOGM Tiles

The created EOGM tiles from multiple intelligent vehicles are uploaded when the GraphSLAM and quad-tree tiling are complete and the network to cloud service is available. The uploaded EOGM tiles are matched to the exiting EOGM tiles in the cloud using the quad-key matching. Then, the uploaded EOGM tiles are merged into the matched EOGM tiles in the cloud. The merged EOGM tiles in the cloud can be downloaded to other intelligent vehicles for the environment modeling and SLAM initialization. There are many benefits when the global EOGM tiles generated by individual vehicles are shared with multiple vehicles via cloud service. First, the intelligent vehicle can obtain the environmental information about the areas of future visits by downloading the global EOGMs of the area from cloud. Second, the GraphSLAM performance of the individual vehicles for the new visited area is improved by initializing the SLAM using the downloaded global EOGMs. Finally, it is possible to obtain the accurate and reliable global EOGMs by integrating the several EOGM tiles from multiple vehicles.

To merge the uploaded EOGM tiles from multiple intelligent vehicles to the existing EOGM tiles in the cloud, Dempster combination rule of evidential theory with aging effect is applied. The aging of the old EOGM tiles in the cloud must be considered for the merging because there is a probability of changing the static environment as time passes. The aging effect can be implemented by decay factor α that can be described by
(14)$$\alpha =exp(\frac{t_{old}-t_{new}}{\tau })$$
where $t_{new}$ is the time of grid map uploaded from the vehicles, $t_{old}$ is the time of grid map stored in the cloud and τ is the time constant that determines the decaying rate. By applying the decay factor α to the EOGM, we can obtain the aging EOGM as
(15)$$\begin{array}{ccc} {}^{\alpha }m_{G\left\{i,j\right\},t_{new}}\left(A\right) & =\alpha \cdot m_{G\left\{i,j\right\},t_{old}}\left(A\right), & A\subset \Omega \\ {}^{\alpha }m_{G\left\{i,j\right\},t_{new}}\left(\Omega \right) & =1-\alpha +\alpha \cdot m_{G\left\{i,j\right\},t_{old}}\left(\Omega \right) \end{array}$$

Then, the global EOGMs in the cloud are updated using the Dempster combination rule:(16)$$m_{G\left\{i,j\right\},t_{new}}{=}^{\alpha }m_{G\left\{i,j\right\},t_{new}}⊕m_{L\left\{i,j\right\},t_{new}}.$$

## 5. Experiments

### 5.1. Experimental Environments

A test vehicle is equipped with six automotive LiDARs (Valeo ScaLa) and a single GPS receiver (NovAtel receiver without any corrections). The six four-layers scanning LiDARs can cover the $360^{\circ }$ Field-of-view (FoV) and, the maximum detection range of LiDAR is 150 m, the horizontal resolution is $0.25^{\circ }$, the vertical resolution is $0.8^{\circ }$, and the distance resolution is 0.1 m. The Cartesian local EOGM covers a distance of 100 m around the vehicle with 0.2 m grid size. The confidence coefficient λ which is represented in (Equation 1) is set to 0.7 with considering of the LiDAR noise characteristics. The test vehicle’s CAN (controller area network) provides the vehicle motion data from the onboard sensors (wheel speed sensors, yaw rate, steering angle sensors, and accelerometer). The GPS data used to initialize and bound the pose nodes of GraphSLAM. An embedded computer (Intel i7-4700EQ CPU 2.4 G) on the test vehicle runs GraphSLAM to create the EOGM for the recording trajectory. The generated EOGM are converted to geodetic quad-tree tile format and uploaded to the map cloud server (Amazon Web Services) via a 4G mobile network.

### 5.2. Tiled EOGM Generation for Large-Scale Area

An experiment was performed to evaluate the large-scale coverage of the quad-tree tiled EOGMs. The test trajectory was about 20 km (Figure 8). The red tile shows a single large tile that covers the entire trajectory, and the blue tiles represent the tiles of the quad-tree level 17. The length of one side of the level 17 tiles is about 250 m. The reason why the blue tiles exceed the red tile at the boundary is that quad-tree tiles have their own global position and size. Figure 9 shows the memory usage of various level tiles for the mapping. The smaller size (higher level) tiles improve the memory efficiency because they effectively place tiles only on trajectories without wasting space. In addition, this memory efficiency is achieved by little computational cost because only calculation of quad-key is required for tile based EOGM mapping.

Since the EOGM tiles have the same data structure as images, it is available to store the global EOGMs into image files. The amount of storage required for the entire level 17 tiles is about 654 megabytes using the bitmap format, but it can be reduced to an average of approximately 1.28 megabytes using the lossless data compression of PNG (portable network graphics). The one file of the compressed EOGM tile is about 150 kilobytes, and this size is acceptable to share with other vehicles via mobile Internet.

In addition to the memory efficiency, the coordinate conversion error from the plane approximation can be reduced by the adaptive reference point of each quad-tree tile. Since the approximation error of the coordinate conversion is difficult to evaluate using quantities value due to the dependence of the evaluation location, we evaluate the approximation performance by analyzing the distance between the reference point (origin) and conversion point. When there is one reference point (start point) for the entire trajectory, the maximum distance from the reference to the vehicle position is about 15 km. However, for the level 17 quad-tree tiles, the maximum distance is reduced to about 350 m. This means that the the quad-tree tile system reduces the approximation error through the mechanism as shown in Figure 5.

### 5.3. Cloud Update of EOGMs

Another experiment was performed to evaluate the EOGM GraphSLAM and the cloud update of EOGM tiles for the multiple mappings. The test site was an urban area where has moving objects and parked cars, as show in Figure 10. Although we had only one test vehicle, the cloud update experiments for the multi-vehicle was available by multiple driving, mapping, and uploading. We performed six tests at different times to create and update the multiple EOGMs, as shown in the mapping trajectories of Figure 10.

The test site can be evenly divided into several tiles of the quad-tree level 17. The tile boundary and tile ID for level 17 tiling of the test site is shown in Figure 10. Each created EOGM tiles for each mapping are uploaded to the cloud and merged into one EOGM tiles, as shown in Figure 11. For the EOGM cloud update, new EOGM tile must be matched with existing tiles in cloud. Since each tile has a unique quad-key, it is straightforward to search and manage the tiles matching using a hash table.

Figure 12 shows the EOGM update over time for the red box area in the Figure 10. The EOGM for the area were updated three times: EOGM mapping 1 (09:12), mapping 2 (09:34), and mapping 5 (15:46). Figure 12a shows the aerial image. Figure 12b shows the EOGM after first cloud update of mapping 1. Although there are many moving vehicles and pedestrians, the only static objects such as parked cars and non-moving cars are mapped. Figure 12c shows the EOGM of mapping 2, and it merged to (b). The merged result of Figure 12b,c is shown in Figure 12d. The static evidence (red in the white box) of a changed objects are blurred due to EOGM merging with taken into account the aging. The time constant τ which determines the decaying rate is 24 h. This value means that the environment which does not change in 24 h can be considered as static environment. After about six hours, EOGM was generated by mapping 5 as shown in Figure 12e. Some parked cars have left and new parked car has appeared. The merged result of Figure 12d,e is shown in Figure 12f. Since six hours have passed, old maps are reflected weakly and new updated maps are more reflected.

## 6. Conclusions

This paper presents a cloud update framework of multi-vehicle EOGMs based on the GraphSLAM, geodetic quad-tree tile system, and the evidential cloud update of EOGM tiles.

(1) The evidential theory for the EOGM provides a basis to manage all types of cell states such as *free*, *occupied*, *unknown*, and *conflict*, which cannot be managed by probabilistic theory. In addition, the Dempster combination rule allows merging the uploaded EOGM tiles from the multiple vehicles to the old EOGM tiles in the cloud.

(2) The geodetic quad-tree tile system provides a common tile interface to manage the EOGMs for the entire world. The sharing of the global EOGMs of multiple intelligent vehicles is based on the geodetic quad-tree tile format. In addition, the management based on the quad-tree tile system provides many benefits, including the improvement of memory utilization, reduction of a coordinate conversion error, and acceleration of searching speed by tree based searching. Since the geodetic tiling method only requires the calculation of quad-key, the computing cost can be ignored.

(3) The proposed algorithm was evaluated through two experiments. The first experiment evaluated the ability of the EOGM tiling to manage large-scale area (more than 20 km). The second experiment verified the cloud update process of multiple EOGM tiles in consideration of aging effect. After uploading the EOGMs from individual vehicles, a post processing and quality evaluation of the uploaded EOGMs are required in cloud. Since the computing time of post processing and evaluation are not fast enough, the real-time sharing is not available currently. For our future work, we have a plan to share the EOGMs in real-time by reducing the computational time of post processing and evaluation. If the post processing technique is mature in the future, real-time download will be possible.

The presented study limits the environment model into 2D plane coordinates because of the computation and memory constraints. Future studies will extend the 2D EOGM to 3D EOVM (evidential occupancy voxel map) with a GPU parallel processing in order to provide a more detail environment model for intelligent vehicles.

## Figures and Tables

**Figure 1 sensors-18-04119-f001:**
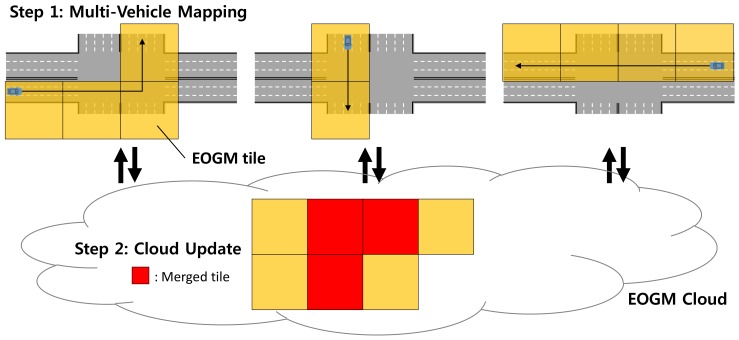
Two-step framework for EOGM cloud update of multi-vehicle mapping.

**Figure 2 sensors-18-04119-f002:**
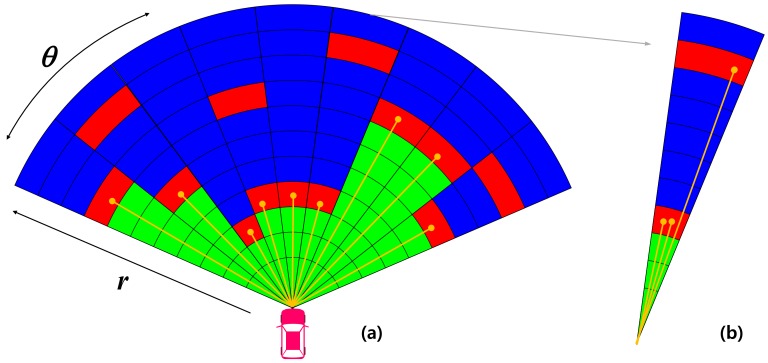
Local EOGM generated from the LiDAR (*F*: green, *O*: red, Ω: blue). (**a**) Local EOGM in polar coordinate, (**b**) Unit polar grid.

**Figure 3 sensors-18-04119-f003:**
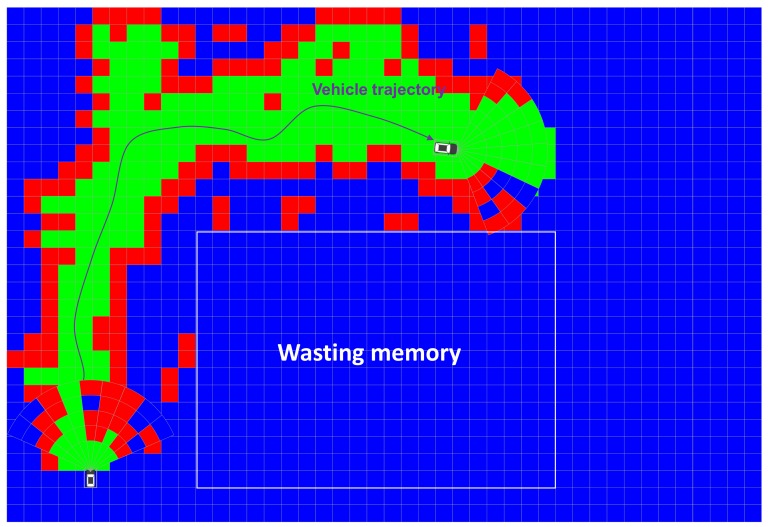
Mapping of Global EOGM based on the merging of local EOGM.

**Figure 4 sensors-18-04119-f004:**
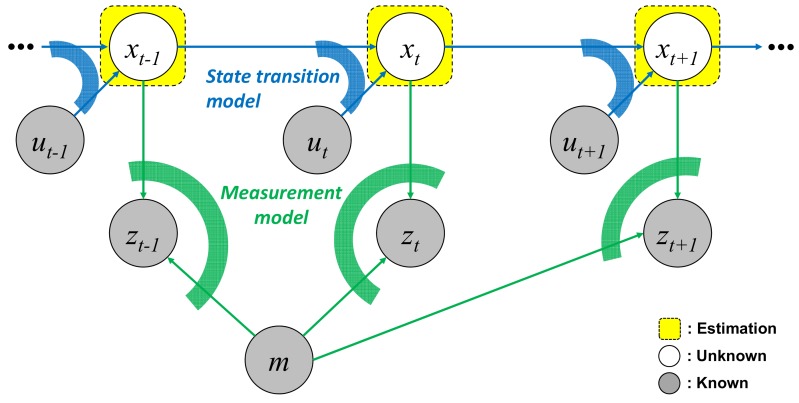
SLAM represented by a directed graph of a dynamic Bayesian network (DBN) based on the assumptions of the Markov process.

**Figure 5 sensors-18-04119-f005:**
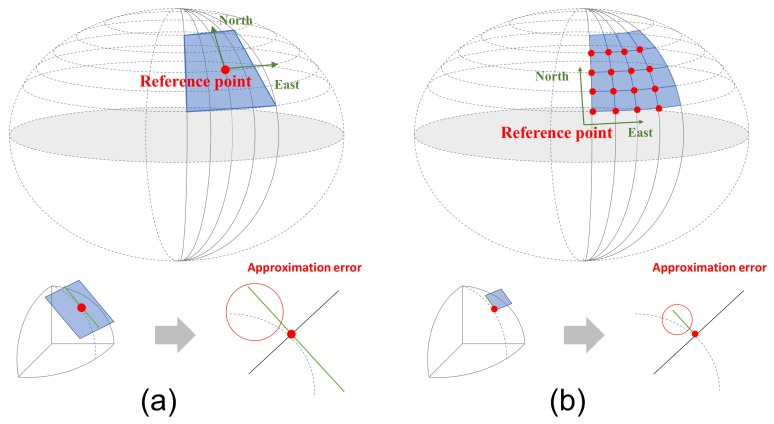
Coordinate conversion between geodetic coordinate and Cartesian coordinate for mapping of global EOGM. Plane approximation (**a**) without the quad-tree tile and (**b**) with the quad-tree tile.

**Figure 6 sensors-18-04119-f006:**
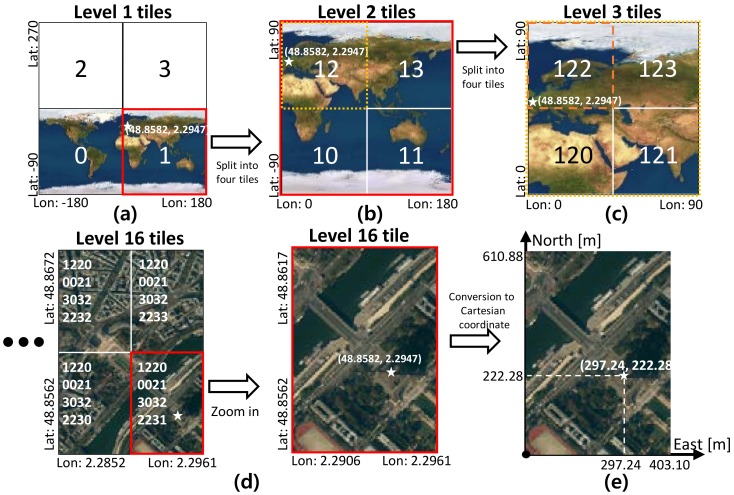
Geodetic quad-tree tile system for global EOGM management. tiles of level 1–16, and tile for global EOGM in Cartesian coordinate.

**Figure 7 sensors-18-04119-f007:**
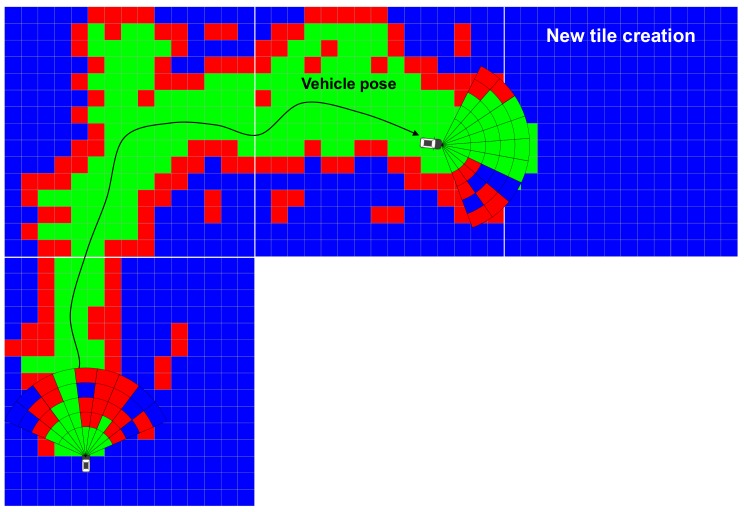
Mapping of Global EOGM based on the quadtree tile.

**Figure 8 sensors-18-04119-f008:**
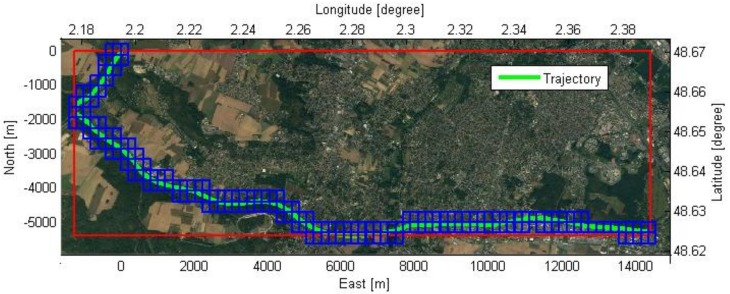
The test trajectory (20 km) to evaluate the large-scale coverage of the geodetic quad-tree EOGM tiles.

**Figure 9 sensors-18-04119-f009:**
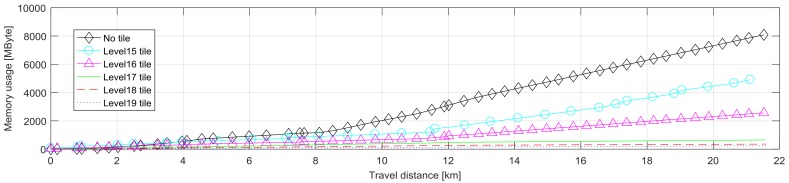
Memory usage for the various level tiles to evaluate the memory efficiency of the quad-tree tile system.

**Figure 10 sensors-18-04119-f010:**
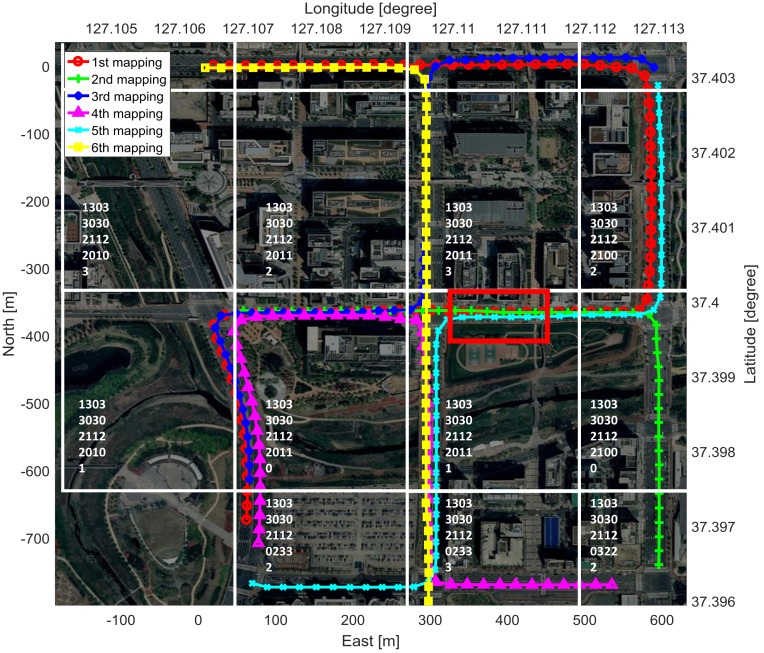
Trajectories and quad-tree tiles for the six test driving at different times in an urban area.

**Figure 11 sensors-18-04119-f011:**
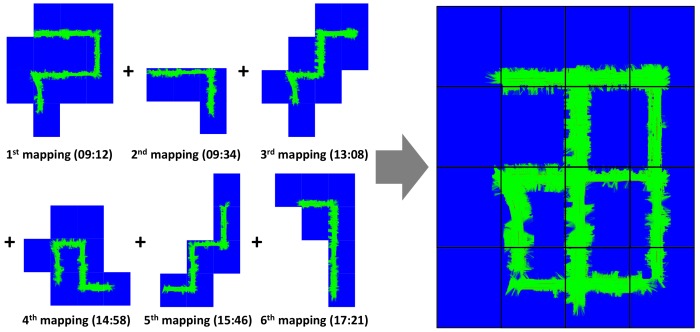
Cloud update of the multiple EOGM tiles for the different areas at different times. EOGM tiles of six test mapping are uploaded and merged into EOGM tiles in the cloud.

**Figure 12 sensors-18-04119-f012:**
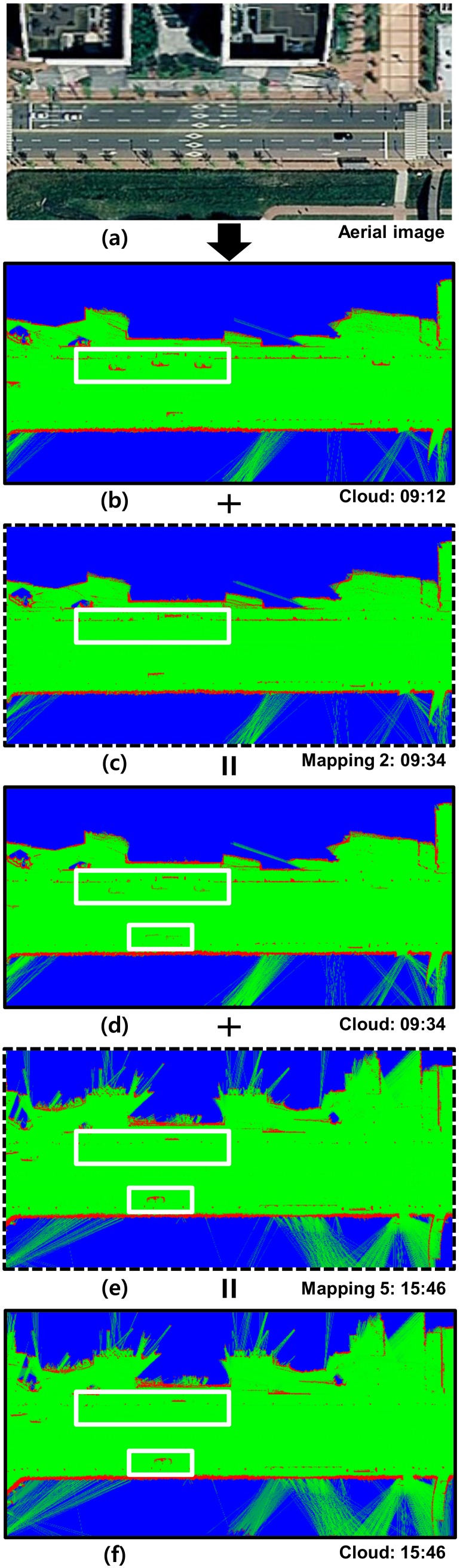
Cloud EOGM update over time in the urban area.

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
