# Peer review of "Cloud Update of Tiled Evidential Occupancy Grid Maps for the Multi-Vehicle Mapping"

_sensors, 2018, doi:10.3390/s18124119_

Reviewer 1 Report

The paper presents a cloud map update from multi-vehicle mapping based on Evidential Occupancy Grid Maps (EOGM). The paper proposes the utilization of Geodetic quad-tree tile system for the map merging from multi-vehicle because the system provides a standard data structure for map merging. Also, it introduces the ageing function for the map merging in the cloud to weaken the inference of old map data. 

Although the used techniques in the paper are conventional (EOGM, GraphSLAM, Dempster-Shafer theory, and Geodetic quad-tree tile system), I found cloud map update based on Geodetic quad-tree tile system is promising because the way is straightforward. 

From page 3 to page 7 describes the existing works including EOGM, Dempster combination rule (conjunctive combination,  Dempster normalization), convert from polar grid to Cartesian grid, GraphSLAM. It is a bit too redundant for the literature. Using the references, it can be minimized to at least half size. It much better to focus on the paper’s contribution rather than the literature description. 

The real-time map sharing is essential to determine the optimal behavior. The paper does not evaluate nor consider how real-time is the system. 

Since the system assumes the incremental upload to the cloud, download frequency is the key for the map receiver. Please add the discussion or evaluation about the realtimeness of the system. If it is not necessary, please describe the reason. 

Minor error: "evidential GraphSLAM" appear in conclusion. Please explain it or fix it. 

Author Response

1.1 From page 3 to page 7 describes the existing works including EOGM, Dempster combination rule (conjunctive combination, Dempster normalization), convert from polar grid to Cartesian grid, GraphSLAM. It is a bit too redundant for the literature. Using the references, it can be minimized to at least half size. It much better to focus on the paper’s contribution rather than the literature description.

Answer: As the reviewer commented, we agree that the EOGM, Dempster theory and coordinate conversion are very conventional methods in SLAM field. However, we think that the readers of this paper should be able to easily implement the proposed method again and evaluate the results of paper. Therefore, the description of the overall processes of this framework (containing sensor model, evidential theory, and GraphSLAM) will help readers to understand the proposed cloud update system.

1.2 The real-time map sharing is essential to determine the optimal behavior. The paper does not evaluate nor consider how real-time is the system. Since the system assumes the incremental upload to the cloud, download frequency is the key for the map receiver. Please add the discussion or evaluation about the realtimeness of the system. If it is not necessary, please describe the reason.

Answer: The proposed algorithm was evaluated through two experiments. The first experiment evaluated the ability of the EOGM tiling to manage large-scale area (more than 20 km). The second experiment verified the cloud update process of multiple EOGM tiles in consideration of aging effect. After uploading the EOGMs from individual vehicles, a post processing and quality evaluation of the uploaded EOGMs are required in cloud. Since the computing time of post processing and evaluation are not fast enough, the real-time sharing is not available currently. For our future work, we have a plan to share the EOGMs in real-time by reducing the computational time of post processing and evaluation. If the post processing technique is mature in the future, real-time download will be possible.

1.3 Minor error: "evidential GraphSLAM" appear in conclusion. Please explain it or fix it.

Answer: This paper presents a cloud update framework of multi-vehicle EOGMs based on the GraphSLAM, geodetic quad-tree tile system, and the evidential cloud update of EOGM tiles.

Reviewer 2 Report

This is a very well-written paper, clearly organized, and complete with state of the art analyses.

Author Response

We thank your sincere comments.

Reviewer 3 Report

Overall, I believe that this work is novel and valuable. However, there are some flaws which need to be enhanced.

1. Related and classical works should be added in the introduction. In this paper, only several general methods are briefly introduced in the introduction. Specific methods and related works should be added to your introduction.

2. All methods you proposed in your paper have been proposed before. What is the main contributions of this paper, and the contributions should be explained clearly in Introduction.

3. Many previous works have been made, so the comparison between your method and previous methods should be made to validate the better performance of your method.

4. Only one set of experiments was carried out in the paper. If possible, it is best to conduct multiple sets of experiments for comparison and analysis.

5. There is only an analysis of memory performance in the chart, and lack of analysis of calculation speed and time latency.

Author Response

3.1 Related and classical works should be added in the introduction. In this paper, only several general methods are briefly introduced in the introduction. Specific methods and related works should be added to your introduction.

Answer: As reviewer commented, there is only general explanation for previous works. We supplemented the specific description of the related works about probabilistic and evidential occupancy grid map.
We added the some specific related works in introduction, as below:

Probabilistic approaches have been applied to infer the occupancy state of grid cells with consideration of sensor noise characteristics [5–7]. These papers provide the probabilistic grid mapping using Bayesian approach using LiDAR beam model. Each cell state of the probabilistic occupancy grid map (POGM) models the posterior probability of occupancy. Although the probabilistic approaches of POGM are able to manage the sensor noise well, there are limits to conditions of sensor blind spots and dynamic objects. For the sensor blind spots, the occupancy state is unknown because of no sensor easurement for the detection area. For the dynamic objects, the occupancy state is conflict because the sensor measurement changes over time. In the probabilistic approach, the unknown and conflict state are only represented by a probability of 0.5, which is not clear to represent the state of the cell.
To overcome the limits of the POGM, evidential (Dempster-Shafer) theory has been applied to model the occupancy grid map [8–10]. The evidential framework can explicitly represent each cell as the occupied, free, unknown and conflict state. There are many previous studies to introduce the Dempster-Shafer approach for modeling of evidential occupancy grid map (EOGM) [11,12]. Karaman, et al. compared the performance of various types of approaches (including Dempster-Shafer approach) for environment modeling, Cao, et al. described the 2D grid map building process using Demspter-Shafer evidence theory. In this literature, simple laser beam model and Demspter-Shafer approach based information fusion for map building were provided. To create the EOGMs for the surrounding environment of the vehicle, SLAM (Simultaneous Localization And Mapping) is applied basically. Celmens, el al. applied the evidential theory to SLAM application. This paper evaluated the evidential theory based SLAM with collision-free path planning. Evidential FastSLAM based on Rao-Brackwellized particle filter was used to build the EOGM and evaluated by comparing with POGM in [14]. Credibilist SLAM simultaneously estimates the vehicle pose and builds the EOGM using LiDAR [15,16]. This paper applies GraphSLAM which is a state-of-the-art method to obtain the pose of a vehicle based on the graph optimization technique [17].

3.2 All methods you proposed in your paper have been proposed before. What is the main contributions of this paper, and the contributions should be explained clearly in Introduction.
Answer:

Contribution
- The contribution is to propose cloud update framework of EOGMs in large-scale real road environment.
Detail explanation
- Although evidential grid map, GraphSLAM and other methods have been proposed before, the proposed cloud update framework for real automotive application and experimental evaluation using these methods might be valuable.
- We specified the contribution in introduction, as below:

Correction
The main contribution of this paper is to propose a cloud update framework of evidential occupancy grid maps (EOGMs) for multiple intelligent vehicles in the large-scale real road environment. Each intelligent vehicle creates the EOGM for their moving trajectory based on GraphSLAM using LiDARs, motion sensors, and low-cost GPS receiver. The created EOGMs are uploaded and merged to EOGM in the map cloud. A standard tiling method of geodetic quad-tree tile system is applied to manage the EOGMs. The tiling method provides a common interface and data format to manage the large-scale EOGMs. Dempster combination rule of the evidential theory provides a theoretical basis to merge the new EOGM tiles to old EOGM tiles with consideration of the aging factor of old tiles. Experiments were performed to evaluate the cloud update framework of multi-vehicle EOGMs in the large-scale area.

3.3 Many previous works have been made, so the comparison between your method and previous methods should be made to validate the better performance of your method.
Answer: One of the main contributions of this paper is to reduce the memory size of EOGMs by applying geodetic tiling system for cloud update framework. In figure 8, 9, the memory size of the proposed framework is compared and evaluated.
3.4 Only one set of experiments was carried out in the paper. If possible, it is best to conduct multiple sets of experiments for comparison and analysis.
Answer: We drove six times for evaluation of the proposed cloud update framework. This experiment was conducted in Pangyo Techno-Valley, Korea which is the complicated urban environment. Since this site contains lots of urban scenarios such as various shapes of buildings, parks, multiple lane roads, and parking lots, the one set of experiments are sufficient to evaluate the proposed framework.
3.5 There is only an analysis of memory performance in the chart, and lack of analysis of calculation speed and time latency.
Answer:Since simple quad-key calculations (such as simple bitwise operation and four fundamental rules of arithmetics) are required for geodetic quad-tree tiling system, the computational cost can be ignorable. We added the description about it in conclusion explicitly, as below
Correction
The smaller size (higher level) tiles improve the memory efficiency because they effectively place tiles only on trajectories without wasting space. In addition, this memory efficiency is achieved by little computational cost because only calculation of quad-key is required for tile based EOGM mapping.
Correction
2) The geodetic quad-tree tile system provides a common tile interface to manage the EOGMs for the entire world. The sharing of the global EOGMs of multiple intelligent vehicles is based on the geodetic quad-tree tile format. In addition, the management based on the quad-tree tile system provides many benefits, including the improvement of memory utilization, reduction of a coordinate conversion error, and acceleration of searching speed by tree based searching. Since the geodetic tiling method only requires the calculation of quad-key, the computing cost can be ignored.
Answer: We added the description about time latency of uploading and downloading for cloud in conclusion.
sensors-389180 Response to reviewers
Page 7 of 7
Correction
3) The proposed algorithm was evaluated through two experiments. The first experiment evaluated the ability of the EOGM tiling to manage large-scale area (more than 20 km). The second experiment verified the cloud update process of multiple EOGM tiles in consideration of aging effect. After uploading the EOGMs from individual vehicles, a post processing and quality evaluation of the uploaded EOGMs are required in cloud. Since the computing time of post processing and evaluation are not fast enough, the real-time sharing is not available currently. For our future work, we have a plan to share the EOGMs in real-time by reducing the computational time of post processing and evaluation. If the post processing technique is mature in the future, real-time download will be possible.